# Neurocognitive Function Domains Are Not Affected in Active Professional Male Footballers, but Attention Deficits and Impairments Are Associated with Concussion

**DOI:** 10.3390/sports12060170

**Published:** 2024-06-20

**Authors:** Lervasen Pillay, Dina Christa Janse van Rensburg, Steve den Hollander, Gopika Ramkilawon, Gino Kerkhoffs, Vincent Gouttebarge

**Affiliations:** 1Amsterdam UMC Location University of Amsterdam, Department of Orthopedic Surgery and Sports Medicine, Meibergdreef 9, 1105AZ Amsterdam, The Netherlands; drpillay@absamail.co.za (L.P.);; 2Section Sports Medicine, Faculty of Health Sciences, University of Pretoria, Pretoria 0028, South Africa; christa.jansevanrensburg@up.ac.za; 3Football Players Worldwide (FIFPRO), 2132LR Hoofddorp, The Netherlands; 4Department of Statistics, University of Pretoria, Pretoria 0028, South Africa; 5Academic Center for Evidence-Based Sports Medicine (ACES), 1105AZ Amsterdam, The Netherlands; 6Amsterdam Collaboration on Health & Safety in Sports (ACHSS), IOC Research Center of Excellence, 1105AZ Amsterdam, The Netherlands; 7Amsterdam Movement Sciences, Aging & Vitality, Musculoskeletal Health, Sports, 1105AZ Amsterdam, The Netherlands

**Keywords:** executive functioning, reaction time, cognitive flexibility, psychomotor speed, complex attention, processing speed, sport, football

## Abstract

Objective: To determine the neurocognitive function of active professional male footballers, determine whether deficits/impairments exist, and investigate the association between previous concussion(s) and neurocognitive function. Methods: An observational cross-sectional study conducted via electronic questionnaires. The CNS Vital Signs online testing system was used to evaluate neurocognitive function. Results: Of the 101 participants, 91 completed the neurocognitive function testing. Neurocognitive function domain deficits or impairments were unlikely in 54.5–89.1%, slight in 5.9–21.8%, moderate in 1.0–9.9%, and likely in 4.0–14.9% of participants. A history of zero concussions found a significant association between the neurocognitive index (Odds Ratio [OR] 0.6; 95% CI 0.2–0.4) and complex attention domain (OR 0.3; 95% CI 0.1–0.9), with 40% and 70% less odds, respectively, of deficit/impairment. Among the 54.5% who reported any number of concussions, there were increased odds of neurocognitive domain deficits/impairments for complex attention (CA) [3.4 times more] and simple attention (SA) [3.1 times more]. Conclusion: In the active professional male footballer, most neurocognitive functions do not have significant deficits/impairments. The odds of neurocognitive function deficit/impairment were significantly increased threefold for CA and SA in those who reported a history of any concussion(s).

## 1. Introduction

According to Collin’s English dictionary, “neurocognitive” relates to cognitive functions (the mental process involved in knowing, learning, and understanding things) associated with particular brain areas [1]. Neurocognitive function is an active and passive brain process that allows one to function in one’s environment. In sports (match or training), athletes need to have the ability to perform a situational analysis using visual, verbal, and physical inputs, be adaptable, process this information for the desired outcome, and act upon the decision [2]. These external factors are especially compounded in team sports such as football, as each team member must make different decisions to achieve the common outcome of scoring a goal. Internal factors, e.g., mental fatigue, can also affect neurocognitive function [3]. These external and internal factors have an interdependent relationship and influence neurocognitive function. Football, like most sports, requires a player to have optimal physical and neurocognitive functions to perform at a professional level. A deficit or impairment in one or more domains of neurocognitive function may be associated with decreased sports performance [2] and an increased risk of musculoskeletal injuries [4].

Most research on neurocognitive function in football focuses on its association with concussion or sub-concussive trauma (repetitive heading) in active and retired players [5,6]. However, concussion as a contributing factor for neurocognitive deficit and impairment has still not been thoroughly established in the research [7,8].

Neurocognitive function deficits and impairment scores are reported in different at-risk populations such as patients with cancer [9], human immunodeficiency viral disease [10], and auto-immune disease [11]) and are compared with the general population normative scores. The literature fails to adequately report on (i) the neurocognitive function scores of active professional male footballers compared with the general population (and if this, indeed, is comparable) and (ii) whether neurocognitive function domain deficits and impairment exist in active professional male footballers. Only one study (among Swedish elite development of male and female football players) reported a difference in normative scores in elite footballers compared with the general population [12]. The lack of research on neurocognitive function in active professional male football players requires redress. A full battery of neurocognitive function evaluation may assist clinicians in determining normative scores of neurocognitive function in professional male football players and identifying deficits and impairments in an improved manner. As this evidence base develops, more attention may then be focused on other aspects of neurocognitive function in professional male football players, including its role in performance and injuries.

In our study, we aimed to expand the body of evidence specific to the active professional male footballer and the aspect of neurocognitive function. Therefore, our first objective was to determine the scores of the overall neurocognitive function and its domains in this cohort and whether deficits or impairments exist. Our second objective was to investigate the association between previous concussion(s) and overall neurocognitive function and its domains.

## 2. Methods

### 2.1. Design

An observational cross-sectional study was conducted, using the Strengthening the Reporting of Observational Studies in Epidemiology statement to guarantee the reporting quality [13]. The Medical Ethics Review Committee of the Amsterdam University Medical Centers (Amsterdam UMC, location AMC) provided ethical approval for the study (Drake Football Study: NL69852.018.19|W19_171#B202169). The study was conducted in accordance with the Declaration of Helsinki (2013).

### 2.2. Participants

The study population consisted of active professional male footballers recruited by Football Players Worldwide (FIFPRO) through affiliated national unions from Europe. Inclusion criteria for participants were that they should be a professional footballer of male gender aged between 24 and 30 years, able to read and comprehend English or French, and not presently being treated for a confirmed or suspected concussion. For our study, we defined a professional male footballer as one who trains to improve performance, competes in the highest or second-highest national league, and has football training and competition as a major activity (way of living) or focus of personal interest, devoting several hours in all or most of the days for these activities and exceeding the time allocated to other types of professional or leisure activities [14]. We relied on Green’s procedure’s general rule of thumb to calculate the sample size [15]. As we required at least 50 participants added to the number of independent variables, we set the intended sample size in our study at a minimum of 58 participants [15].

### 2.3. Concussions (Independent Variable)

A single question examined the history of football-related concussions sustained during training and/or matches [‘How many concussions have you had so far during your professional football career (training and competition)?’]. Participants were requested to consult their medical record or team physician to answer this question. We defined concussion as a blow (direct or transmitted) to the head resulting in clinical, neurological, and cognitive symptoms for this study [16]. This definition was clearly stated to the participants in the questionnaire.

### 2.4. Neurocognitive Functions (Dependent Variables)

Several domains of neurocognitive function were assessed using the neuropsychological CNS Vital Signs (CNS-VS) online testing system (CNS Vital Signs LLC, Morrisville, NC, USA; https://www.cnsvs.com) on a desktop or laptop [17]. The CNS-VS online testing system is available in several languages and has been previously used in various professional sports, including boxing, football, and rugby [12,18,19,20]. The psychometric properties of the CNS-VS tests are similar to conventional neurocognitive testing, with a moderate to good level of reliability (test–retest correlation coefficients: 0.65–0.88) and validity (concurrent validity correlation coefficients: up to 0.79) [21,22]. The CNS-VS tests can also differentiate between healthy subjects and those with various psychological or neurological disorders or diseases [23,24]. The CNS-VS program outputs participant scores (raw scores) and their corresponding standard scores. Standard scores [a mean of 100 and standard deviation (SD) of 15] are calculated from the raw scores against age-matched normative scores based on the United States general population. Owing to the high variability of raw scores, we used the standard scores for clinical interpretation [21]. The program has embedded processes that determine if the participant is manipulating testing performance for secondary gain or simply misunderstood the testing procedure or process, thereby validating the scores. In our analysis, we only used scores determined as valid by the program. Ten neurocognitive function domains (verbal memory, visual memory, psychomotor speed [PM Spd.], reaction time [RT], complex attention [CA], cognitive flexibility [CF], processing speed [PS], executive function [EF], simple attention [SA], and motor speed [MS]) were assessed through the seven CNS-VS subtests described below. The program also calculates a Neurocognitive Index (NCI) score, which is the overall neurocognitive function score of the participant for all the domains being tested. The lower the scores, the more increased the likelihood of neurocognitive function domain deficit or impairment. Participants were grouped (as defined by the program guidelines) for NCI and neurocognitive function domains according to their valid standard scores (*unlikely* neurocognitive deficit or impairment, >90; *slight* neurocognitive deficit or impairment, 80–90; *moderate* neurocognitive deficit or impairment, 70–79; and *likely* neurocognitive deficit or impairment, <70).

#### 2.4.1. Composite Memory

The Composite Memory (CM) domain combines the scores of the Verbal Memory and Visual Memory domains of the test.

#### 2.4.2. Verbal Memory Test

The Verbal Memory test (VBM) measures how well a participant can recognize, remember, and retrieve words. For this test, participants must remember 15 words that appear one at a time on the screen every two seconds. Then, the participant has to identify those same words nested among 15 new words (immediate recognition), and a delayed trial is conducted at the end of the test (delayed recognition). Low scores indicate verbal memory impairment.

#### 2.4.3. Visual Memory Test

The Visual Memory test (VSM) measures how well a participant can recognize, remember, and retrieve geometric figures. For this test, participants must remember 15 geometric figures that appear one at a time on the screen every two seconds. Then, the participant has to identify those same geometric figures nested among 15 new geometric figures (immediate recognition). A delayed trial is conducted at the end of the test (delayed recognition). Low scores indicate visual memory impairment.

#### 2.4.4. Finger Tapping Test

The Finger Tapping test (FTT) measures motor speed and fine motor control. After a practice round, three rounds require the participant to press the Space Bar with their right index finger as many times as possible in 10 s. The test is repeated with the left index finger. Low scores indicate motor slowing.

#### 2.4.5. Symbol Digit Coding Test

The Symbol Digit Coding test (SDC) measures visual processing and requires several simultaneous cognitive functions (visual scanning, perception, memory, and motor function). The test consists of serial presentations of screens, each containing boxes with eight symbols above and eight associated numbers below. The participant must type the associated number for each symbol, randomly presented in a test box. Errors may be due to misperception, confusion, or impulsive responses.

#### 2.4.6. Stroop Test

The Stroop test (ST) measures whether a participant can adapt to a rapidly changing and increasingly complex set of instructions by assessing simple and complex reaction time, inhibition/disinhibition, mental flexibility, and directed attention. It consists of three components. In the first component, the words RED, YELLOW, BLUE, and GREEN appear randomly in black on the screen. The participant presses the space bar as soon as they see the word. In the second component, the words RED, YELLOW, BLUE, and GREEN appear in color on the screen. The participant presses the space bar when the color of the word matches the word. In the third component, the words RED, YELLOW, BLUE, and GREEN appear in color on the screen. The participant presses the space bar when the color of the word does not match the word. Prolonged reaction time indicates cognitive impairment, while errors may be due to misperception, confusion, or impulsiveness.

#### 2.4.7. Shifting Attention Test

The Shifting Attention test (SAT) measures the executive functioning of a participant (the ability to shift from one set of instructions to another quickly and accurately) by using geometric figures with different shapes and colors and matching either the shape or color. The best responses are several correct responses with few errors and short response times.

#### 2.4.8. Continuous Performance Test

The Continuous Performance test (CPT) measures sustained attention and attention over time. The participant is asked to press the space bar when the letter “B” appears on the screen but not any other letter. Long response times indicate cognitive impairment or slowness and may indicate attention dysfunction.

### 2.5. Procedures

Information about the study was sent via email to potential participants by FIFPRO and affiliated national unions, and email addresses were hidden from the principal researcher for privacy reasons. Interested participants gave informed consent and completed an electronic questionnaire (CastorEDC, CIWIT B.V, Amsterdam, the Netherlands) available in English and French. The electronic questionnaire included questions about concussion and the following descriptive variables: age, field position, level of football, number of seasons as a professional football player, family history of diagnosed neurological disease, and self-reported global physical health (using the Patient-Reported Outcomes Measurement Information System Global Health short form; PROMIS-GH) [25]. We did not collect information relating to the timing of data collection after injury, injury severity, recovery, or other confounding medical-related history (e.g., hospitalization due to illness), as this was beyond the scope of this study. Participants were subsequently requested to perform the seven CNS-VS tests. The responses to the questionnaires and tests were coded and made anonymous for privacy and confidentiality reasons. Once completed, the electronic questionnaires and tests were saved automatically on a secured electronic server that only the principal researcher could access. Players participated voluntarily in the study and were not rewarded for their participation. All data were collected between January 2020 and April 2021.

### 2.6. Statistical Analysis

For this study, IBM SPSS version 26 was used for statistical analysis. Descriptive analysis was conducted on all demographic information collected and included frequency, mean, standard deviation (SD), and percentages.

For our first aim, descriptive analysis was performed and presented as percentages and SD for each neurocognitive function domain. Firstly, the percentage of valid responses and secondly, the percentage of participants that were unlikely, slight, moderate, and likely to have neurocognitive function domain deficits or impairments. These deficits and impairments were only calculated in those CNS-VS tests that were valid.

For our second aim, we used logistical regression to determine the odds of developing neurocognitive dysfunction after any, one, two, or three or more concussions. This was presented as odds ratios at a 95% confidence interval (CI). Only the data of those participants who performed valid CNS-VS tests were analyzed.

## 3. Results

### 3.1. Participant Demographics

Consent to participate was obtained from 101 participants. The demographic questionnaires were completed by all consenting participants (Table 1). The mean age of the participants was 26.5 years, and 88.1% (*n* = 89) played at the highest or second-highest national level. The mean PROMIS-GH Physical Health and the Mental Health T-scores were suggestive that active professional male footballers’ physical and mental health was similar to the average United States reference population. While participants did not report being diagnosed with a neurological disease, 21.8% (*n* = 22) reported a family member being diagnosed with a neurological disease.

### 3.2. Neurocognitive Functions

Of the 101 participants, 90% (*n* = 91) performed the CNS-VS tests. Valid tests for each neurocognitive function domain ranged between 77.2% and 90.1%. NCI scores were valid in 77.2% of the participants, with a mean standard score of 86.47 (SD = 31,07). Of the valid NCI scores, 77.2% were unlikely to have neurocognitive function domain deficits or impairments. The remaining 22.8% had a slight to likely possibility of having neurocognitive function domain deficits and impairments. Specific neurocognitive function domain deficits or impairments were *unlikely* in 54.5% to 89.1%, *slight* in 5.9% to 21.8%, *moderate* in 1.0% to 9.9%, and *likely* in 4.0% to 14.9% of participants. Further results are reported in Table 2.

### 3.3. Number of Concussions

A total of 54.5% (*n* = 55) of participants reported that they had sustained at least one concussion during their careers thus far. Of these, one concussion was reported by 23.8% (*n* = 24), two by 18.8% (*n* = 19), and three or more by 11.9% (*n* = 12). Concussions were reported by 16.8% (*n* = 17) of goalkeepers, 20.8% (*n* = 21) of defenders, 10.9% (*n* = 11) of midfielders, and 6% (*n* = 6) of forwards.

### 3.4. Association between Concussion and Neurocognitive Function 

Details of these associations are reported in Table 3. In those participants with no history of concussion, a significant association was found in NCI (OR 0.6; 95% CI 0.2–0.4) and CA (OR 0.3; 95% CI 0.1–0.9), with 40% and 70% less odds, respectively, of having any deficits and impairments in these particular neurocognitive function domains compared with those that reported any concussion. There was a significant association between those who reported any number of concussions and the odds of neurocognitive domain deficits and impairments for CA, which increased by 3.4 times (OR 3.4; 95% CI 1.1–10.1) and SA by 3.1 times (OR 3.1; 95% CI 1.0–9.3) compared with those who did not report a history of concussion. Where a history of any number of concussions was reported, the odds of neurocognitive function domain deficits and impairments were increased by 1.8 times for NCI (OR 1.8; 95% CI 0.8–4.7) and CF (OR 1.8; 95% CI 0.7–4.3) and 1.9 times for EF (OR 1.9; 95% CI 0.8–4.6) compared with those who reported no history of concussion. In those participants who reported a history of one, two, or three or more concussions, there was an increase in the odds of deficits or impairments of five different neurocognitive function domains spread across the concussion variables ranging between 1.2 (OR 1.2; 95% CI 0.3–3.3) and 3.5 (OR 3.5; 95% CI 0.9–3.6) times compared with those who reported no history of concussion.

## 4. Discussion

Our study reported that most neurocognitive function domains do not have significant potential deficits or impairments in the active professional male footballer. Among the more than half of active professional male footballers who reported a history of concussion(s), one in five were defenders. The odds of deficits and impairments were significantly increased by threefold in the domains of CA and SA compared with those players who have never reported a concussion/s.

### 4.1. Neurocognitive Function

Among the 101 participants, reports of a first and second family member history of diagnosis of Parkinson’s disease (9.9%) and Dementia (including Alzheimer’s disease) (11.8%) were significantly higher than the prevalence of Parkinson’s disease (0.005–0.3%) [26] and Dementia (1–7%) [27] in Europe, suggesting an increased genetic predisposition to develop neurocognitive deficits and impairments. Since normative neurocognitive function scores have not been established for active professional male footballers [28], our results were compared with the reference general population. Results of our study reported that seven (CM, PM Spd, RT, CF, ES, SA, and MF) of the 10 neurocognitive function domains had average standard scores similar to the general population, suggestive of normal function at normal capacity. Our findings were similar to those from a previous observational cross-sectional study among top-level Swedish football players (male and female) using the CNS-VS program. The Swedish study reported that apart from the PM Spd domain, all other neurocognitive function domain standard scores were similar to the general population in the group aged 20–29 years at normal function and capacity [12]. In our study, our age inclusion criteria were similar (24–30 years old), but our participant selection was wider (from across Europe), allowing the possibility of generalizing our findings across Europe. In addition, we found that the NCI, CA, and PS standard scores suggested a *slight* to *likely* severity of neurocognitive function deficits and impairments in these domains. The clinical context (whether it affects their daily living activity or function as an active professional male footballer) of these findings should alert clinicians to consider further psychometric evaluation.

### 4.2. Association between Concussion and Neurocognitive Function

A systematic review reported that multiple concussions among the athletic population (mostly American football, soccer, and boxing) appear to be a risk factor for neurocognitive function domain deficits and impairment [29]. Our results do not agree with these findings but rather support the findings of other studies where an association between concussion and effects on neurocognitive function has not been established [7,8,8,30,31,32]. Our study found that although those participants who reported a history of one, two, or three or more concussions had increased odds of deficits and impairments of neurocognitive function in almost all domains, these associations were not significant. We found that when a history of any concussion was reported, there was a significant threefold increase in the odds of having neurocognitive function deficits and impairments of the CA and SA domains. A study among adolescent (male *n* = 36; female *n* = 4) athletes (football *n* = 30 of total participants *n* = 40) with a history of concussion found that EF but not Attention, was negatively affected in the cohort [33]. A systematic review looked at the long-term cognitive outcomes in retired athletes with a history of sports-related concussions. The authors concluded that the neurocognitive function domains often affected are Memory, EF, and Psycho motor function. However, the evidence is weak to have a stance on a cause-effect association between concussion and neurocognitive function in retired athletes [34]. Our study findings, as with other findings, suggest that the effect on neurocognitive function domains in active professional male footballers is variable after reporting concussion(s). Where the odds are increased, clinicians need to consider them in a clinical context.

### 4.3. Clinical Implications

Our study found that lower standard scores for CA and SA were more significantly associated with a history of concussion than no concussion at all. Since Attention is an important aspect of the neurocognitive function domain to be considered in the active professional male footballer, one can argue for neurocognitive function testing on active professional male footballers. Any identified neurocognitive function domain deficits and impairments can be addressed to possibly reduce the risk of injuries [35,36,37] and improve positional play performance [38]. Regular neurocognitive function assessment can be monitored over time, and decreasing scores in certain domains may suggest neurocognitive decline, whether related to concussion(s), disease (e.g., multiple sclerosis), or advancing age. These neurocognitive function deficits and impairments can then be addressed early, utilizing the necessary evaluations (psychological, neurological, or psychiatric review) and interventions (medication or cognitive behavioral therapy).

### 4.4. Research Implications

Neurocognitive function science is evolving in the sporting fraternity. Our study helped to identify research gaps in the literature. Research should be directed to develop normative scores in different athletic populations (football, rugby, American football, Australian rules football, cricket, etc.). Furthermore, a neurocognitive tool common to sports should be developed and utilized to allow for aggregation of analysis, so more concrete recommendations can be established on identifying common neurocognitive function domain deficits and impairments in different sports. More research is required within the active male professional footballer environment in this cohort. It should be monitored over time to determine whether certain neurocognitive function domain deficits and impairments occur as retirement approaches and after retirement. This evidence may assist in identifying those at-risk players and implementing early management of neurocognitive function domain deficits and impairments. This may reduce the incidence of further neurocognitive function decline later in life.

### 4.5. Study Limitations

The study’s ethical approval only allowed for players who were members of FIFPRO to participate in the study. We only collected data from European-based clubs, which may not give a worldwide representative sample among other continental football confederations. A larger sample size may have produced different significant association results between concussions and neurocognitive function domains, as the reporting of concussions is reliant on physician recognition or player reporting. This study forms part of a larger group of studies to inform on the findings among a group of active professional male footballers over the next 10 years. Different participant selections may be spread differently for positional play and may under-report (more forwards with fewer concussions) or over-report (more defenders with more concussions). Even though the CNS-VS program has been validated, it is not commonly used among athletes compared with other computerized neurocognitive function programs, so neurocognitive function domain scores may have been over-reported owing to the thorough nature of the test batteries. Exploring the neurocognitive effect after sustaining injuries, prolonged rehabilitation, and hospitalization due to illness was not explored, as this was not part of this study’s aims. Future studies may consider exploring this aspect.

### 4.6. Study Strengths

This study provides baseline data from this cohort of a sample size that allow for statistically significant interpretation from European players as participants. These data will, in turn, inform future studies that will use the same testing program (e.g., CNS-VS). The cohort was a very specific age group and can be re-interpreted for future study findings. The study only includes male participants—so this study can inform other studies to develop a normative neurocognitive score in this cohort. 

## 5. Conclusions

The results of our study indicate that most neurocognitive function domains do not have significant deficits or impairments in the active professional male footballer compared with the general population. The odds of deficits and impairments were significantly increased by threefold in the domains of CA and SA in those who report any concussion(s) compared with those who have never reported a concussion(s). The clinical impact following the increased odds in the likelihood of neurocognitive deficit or impairment must always be considered. Neurocognitive function should be monitored over time to determine the development of deficits or impairments.

## Figures and Tables

**Table 1 sports-12-00170-t001:** Demographics and characteristics of participants (*n* = 101).

**Demographics**	**Age (Mean and SD)**	**26.5**	1.7
**Football**	**Seasons played (Mean and SD)**	7.6	2.6
**characteristics**	**Playing position (n and %)**		
	Goalkeeper	23	22.8
	Defender	42	41.6
	Midfielder	25	24.8
	Forward	11	10.9
	**Career Level (n and %)**		
	Highest national level	57	56.4
	Second highest national level	32	31.7
	Other levels	12	11.9
**PROMIS-GH**	**Physical Health T-score**	52.9	6.4
**T-scores (Mean and SD)**	**Mental Health T-score**	53.2	7.4
**Neurological**	**Diagnosed player**	0	0
**Disease (n and %)**	**Diagnosed family member**	22	21.8
	**Dementia**	6	5.9
	**Parkinson’s**	10	9.9
	**Alzheimer’s**	6	5.9

n, Number; %, Percentage; SD, Standard Deviation.

**Table 2 sports-12-00170-t002:** Neurocognitive domain function validity, deficits, and impairment.

	Valid %	Unlikely %	Slight %	Moderate %	Likely %	SS Mean (SD)
**NCI**	77.2	77.2	6.9	3.0	13.9	86.47 (31.07)
**CM**	90.1	67.3	14.9	7.9	9.9	**93.64 (20.39)**
**PM Spd**	85.1	75.2	10.9	4.0	9.9	**95.40 (28.40)**
**RT**	86.1	59.4	21.8	9.9	9.9	**90.00 (17.53)**
**CA**	81.2	79.2	8.9	2.0	9.9	81.38 (81.91)
**CF**	86.1	72.3	12.9	7.9	6.9	**92.52 (20.70)**
**PS**	86.1	54.5	20.8	9.9	14.9	84.78 (23.20)
**EF**	89.1	71.3	13.9	8.9	6.9	**93.92 (18.41)**
**SA**	84.2	80.2	7.9	3.0	8.9	**96.98 (18.09)**
**MS**	89.1	89.1	5.9	1.0	4.0	**107.46 (21.23)**

%, Percentage; SD, standard deviation; NCI, Neurocognitive index; CM, Composite Memory; PM Spd, Psychomotor Speed; RT, Reaction Time; CA, Complex Attention; CF, Cognitive Flexibility; PS, Processing Speed; EF, Executive Functioning; SA, Simple Attention; MS, Motor Speed; SS, Standard Score. Note—bold = SS > SS 90 (unlikely deficit or impairment).

**Table 3 sports-12-00170-t003:** Odds ratios of concussions for neurocognitive function domain deficits and impairments.

Concussions					
	0	Any	1	2	>3
	OR (95%CI)	OR (95%CI)	OR (95%CI)	OR (95%CI)	OR (95%CI)
**NCI**	**0.6 (0.2–0.4)**	1.8 (0.8–4.7)	*1.2 (0.4–3.3)*	*2.4 (0.8–7.0)*	0.6 (0.1–2.7)
**CM**	1.0 (0.4–2.3)	1.0 (0.4–2.3)	1.0 (0.4–2.7)	0.5 (0.1–1.5)	*2.3 (0.7–8.0)*
**PM Spd**	1.8 (0.7–4.4)	0.6 (0.2–1.4)	0.8 (0.2–2.2)	0.8 (0.2–2.4)	0.6 (0.1–2.4)
**RT**	1.7 (0.8–3.9)	0.6 (0.3–1.3)	0.5 (0.2–1.4)	*1.8 (0.7–5.1)*	0.3 (0.0–1.0)
**CA**	**0.3 (0.1–0.9)**	**3.4 (1.1–10.1)**	*1.4 (0.4–4.0)*	*2.1 (0.6–6.2)*	*2.1 (0.5–7.6)*
**CF**	0.6 (0.2–1.4)	1.8 (0.7–4.3)	0.8 (0.3–2.3)	*1.7 (0.6–4.8)*	*2.1 (0.6–7.1)*
**PS**	1.0 (0.5–2.2)	1.0 (0.5–2.2)	*1.3 (0.5–3.2)*	*1.4 (0.5–3.9)*	0.4 (0.1–1.3)
**EF**	0.5 (0.2–1.3)	1.9 (0.8–4.6)	0.8 (0.3–2.1)	*2.1 (0.7–6.0)*	*1.9 (0.5–6.7)*
**SA**	0.3 (0.1–1.0)	**3.1 (1.0–9.3)**	*2.0 (0.7–5.8)*	0.7 (0.2–2.5)	*3.5 (0.9–2.6)*
**MS**	1.0 (0.3–3.5)	1.0 (0.3–3.5)	*2.0 (0.5–7.3)*	0.4 (0.0–2.3)	0.7 (0.0–4.3)

NCI, Neurocognitive index; CM, Composite Memory; PM Spd, Psychomotor Speed; RT, Reaction Time; CA, Complex Attention; CF, Cognitive Flexibility; PS, Processing Speed; EF, Executive Functioning; SA, Simple Attention; MS, Motor Speed; OR, Odds ratio; CI, confidence interval. Note: bold text = significant association, italic text = insignificant increased odds.

## Data Availability

All data analyses of data collected are presented in the manuscript. Raw data can be requested from the corresponding author and will be shared at reasonable request.

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
