# Peer review of "Neurocognitive Function Domains Are Not Affected in Active Professional Male Footballers, but Attention Deficits and Impairments Are Associated with Concussion"

_sports, 2024, doi:10.3390/sports12060170_

Round 1
Reviewer 1 Report
Comments and Suggestions for Authors
Dear authors.
Congrats for the research titled “Active professional male footballers do not have significant neurocognitive function domains affected, but Attention deficits and impairments are associated with concussion/s”. It is an interesting research into the analysis of the footballer’s health. However, it is necessary to take into account some recommendations so that the manuscript can be improved:
Abstract:
It is ok.
Introduction:
It is ok.
Materials and Methods:
2.5. Procedures
Include the time that the questionnaire was accessible to the players and from when to when responses to the questionnaire were accepted.
The last paragraph of the Material and Method section (L 205) should be Data analysis (subsection). Here you must explain the statistical analyzes carried out to analyze the results, the statistical program used, the procedure to carry out the analysis, the significance value considered, the normality of the data, etc.
Results:
3.4. Association between concussion and neurocognitive function: The association between concussion and neurocognitive function must be previously explained in the Material and Methods section. In this subsection (3.4.) data related to the odds ratio is presented, however no relationship or association between variables is presented. Explain more.
Discussion:
This section is well discussed.
References:
All cites in the text should be reviewed. They do not correspond to the proper format of the magazine.
It is recommended to review the text since there are some spelling mistakes that must be corrected
With the application of these changes, the quality of the manuscript will be improved.
Thank you
Author Response
Thank you for the reviewer. Please find response point-by-point to your review

Reviewer 2 Report
Comments and Suggestions for Authors
Thank you for the opportunity to review your manuscript, “Active professional male footballers do not have significant neurocognitive function domains affected, but Attention deficits and impairments are associated with concussion/s”
This study aimed to determine global neurocognitive function scores and domains in a cohort and whether deficits or impairments exist. Furthermore, to investigate the association between previous concussions and global neurocognitive function and domains.
The title reads more like a conclusion than an article title. It should be rephrased.
The abstract exceeds the 200 words proposed by the journal. It should be reworded.
Line 69 “The literature fails to adequately report on (1) the…” This reference is inappropriate
Also, check that there are different citation formats in the manuscript.
Line 103-105. A claim is made based on a sample size calculation that is neither named nor justified.
Tables should be placed consecutively from the quoted text.
Author Response
Thank you for the review. Please find a point-by-point response to your review attached

Reviewer 3 Report
Comments and Suggestions for Authors
I believe your topic is of importance. However, as presented, it was difficult to review your manuscript. My hope is my comments are helpful.
Title
There must be a way to reduce your title to 20 words. As opposed to a title, it is a long sentence.
I think you can write concussion and leave off the /s.
Abstract
I counted 338 words. This seems 100 words too long. The results are too comprehensive. Please reduce your abstract to a maximum of 250 words.
7,2% - 90,1% - my guess is you will need to change all , to . 7.2% and so on.
Keywords
Perhaps add athletes or competitors or sport.
Introduction
Line 50, I do not think you need to use (i) and so on given you wrote a sentence with commas and an and.
Line 74, The lack of research on neurocognitive function in active professional male football players requires redress.
I do not understand redress.
Your introduction seems brief and not comprehensive enough.
Methods
As with my statement about line 50, I do not think you need the ( ).
(a) a professional footballer; (b) of male gender; (c) aged between 24 and 30 years; (d) able to read and comprehend English or French; and (e) not presently being treated for a confirmed or suspected concussion. For our study, we defined a professional male footballer as one who (i) trains to improve performance, (ii) competes in the highest or second-highest national league, and (iii) has football training
2.4., please rewrite this section and use 2.4.1 Title and more to make it readable.
Perhaps just by 101 as opposed to n=101, Consent to participate was obtained by n=101 participants
Results
Informative. The tables seem accurate. It is lots of information to digest. Reducing the number of sentences seems a good idea as the information and numbers are found in the tables. I think you can reduce the number of sentences by providing a summary of each test/result. If the numbers are found in the table, then you can just say see the table. The information is repeated too many times.
Discussion
Your titles and subtitles are mixed in format.
Your discussion is very long compared to your very short introduction.
What are the larger group of studies? How much of this study is part of the larger group? This study forms part of a larger group of studies to inform on the findings amongst a group of active professional male footballers over the next 10 years.
Overall, your manuscript needs a thorough edit and formatted to the MDPI system. The references need to be formatted. All the inconsistencies are distracting when trying to provide a review.
I think your topic is important and my suggestions are reasonable.
Comments on the Quality of English LanguageModerate editing required.
Author Response
Thank you for your review. Please find attached a point-by-point response to your review

Reviewer 4 Report
Comments and Suggestions for Authors
The idea of your study is interesting, my recommendations are the following:
I recommend that in the Participants section, the number of study participants and their average age should be mentioned.
Mentioned in section 2.5. Procedures that demographic information was collected regarding: , height, weight, level of education, study activity, but in the Participants section or in table 1 these are not mentioned. I recommend clarification.
I recommend that you clarify in the results section and discuss the total number of subjects who completed the Neurocognitive function part of the notebook, because when 101 appears, when 91 appears.
Author Response

(The authors gave the same response as above.)

Round 2
Reviewer 1 Report
Comments and Suggestions for Authors
Dear authors.
Thank you for considering my suggestions
Reviewer 3 Report
Comments and Suggestions for Authors
Thank you for your revised manuscript and attention to my suggestions.
Comments on the Quality of English LanguageMinor edits